# Allelopathic Potential of *Artemisia absinthium* and *Artemisia vulgaris* from Serbia: Chemical Composition and Bioactivity on Weeds

**DOI:** 10.3390/plants14111663

**Published:** 2025-05-30

**Authors:** Teodora Tojić, Tijana Đorđević, Rada Đurović-Pejčev, Milica Aćimović, Dragana Božić, Ljiljana Radivojević, Marija Sarić-Krsmanović, Sava Vrbničanin

**Affiliations:** 1Faculty of Agriculture, University of Belgrade, Nemanjina 6, 11080 Belgrade, Serbia; teodora.tojic@agrif.bg.ac.rs (T.T.); dbozic@agrif.bg.ac.rs (D.B.); 2Institute of Pesticides and Environmental Protection, Banatska 31b, 11080 Belgrade, Serbia; tijana.djordjevic@pesting.org.rs (T.Đ.); rada.djurovic@pesting.org.rs (R.Đ.-P.); ljiljana.radivojevic@pesting.org.rs (L.R.); 3Institute of Field and Vegetable Crops Novi Sad—National Institute of the Republic of Serbia, Maksima Gorkog 30, 21000 Novi Sad, Serbia; milica.acimovic@ifvcns.ns.ac.rs

**Keywords:** *Artemisia absinthium*, *Artemisia vulgaris*, plant extract, essential oil, allelopathic potential

## Abstract

The use of *Artemisia* species’ plant extracts and essential oils, which are rich in bioactive compounds (allelochemicals), could support weed management. This study focused on the chemical analysis and evaluation of the allelopathic potential of plant extracts (PEs) and essential oils (EOs) of *Artemisia absinthium* and *A. vulgaris* on the germination and early seedling growth of weeds (*Amaranthus retroflexus* and *Setaria viridis*) *in vitro*. The plant extract from *A. vulgaris* showed higher antioxidant activity (IC_50_ = 0.171 ± 0.01 mg/mL) and phenolic content than that from *A. absinthium* (IC_50_ = 0.263 ± 0.01 mg/mL). Chlorogenic acid was the most abundant phenol in both extracts. However, *A. absinthium* contained a higher amount (1.694 ± 0.081 mg/g) and exhibited a stronger inhibitory effect on the germination of *A. retroflexus* (EC_50_ = 0.54 ± 0.02%) and *S. viridis* (EC_50_ = 1.51 ± 0.07%) compared to *A. vulgaris*. The dominant components of *A. absinthium* essential oil were β-thujone (18.9%), cis-ocimene epoxide (7.88%), and bicyclogermacrene (7.04%), while the main constituents of *A. vulgaris* essential oil included gurjunene (10.41%), cis-crysanthenyl acetate (7.17%), and γ-humulene (6.67%). The lowest EC_50_ values for *A. absinthium* essential oil regarding seed germination and seedling length were estimated for *S. viridis* (0.28 ± 0.48% and 0.03 ± 0.00%, respectively), whereas *A. retroflexus* was the most sensitive to *A. vulgaris* essential oil (0.11 ± 0.04% and 0.02 ± 0.00%, respectively). All tested extracts showed allelopathic potential; however, the results indicate that the essential oils had a stronger inhibitory effect than the plant extracts.

## 1. Introduction

Sustainable environmental and production management in modern agriculture means facing up to the challenges of climate change, environmental pollution, depletion of natural resources, and dependence on agricultural inputs. The harmful effects of herbicides on the environment and human health, the ever-increasing number of herbicide-resistant weed populations, the increasing number of invasive alien weed species, the slower development of novel herbicides, the ban on the use of many herbicides, and the intensified focus on organic farming are some of the main factors that have promoted eco-friendly approaches to weed control in recent decades [1,2,3]. In the last two decades, the concept of allelopathy has been employed to reduce our heavy reliance on synthetic herbicides and to find a promising solution to the problems of environmental pollution and herbicide resistance as well as ecological weed management [4,5]. The phytotoxic properties of allelochemicals exuded by allelopathic plants mean they can be a source for the identification and isolation of a wide selection of potential new environmentally acceptable bioherbicides. More than 2000 plant species (39 families) have been found to have strong allelopathic potential [6], but only 3% of the approximately 400,000 known compounds in plants have herbicidal impact [7]. These allelochemicals can influence the germination and growth of weeds through different modes of action [7]. The allelochemicals can affect vital biochemical and physiological processes in plants, e.g., respiration, photosynthesis, cell division and elongation, membrane permeability, water balance, protein biosynthesis, and the activity of many enzymes [8].

Allelochemicals belong to various chemical families, including phenols, flavonoids, terpenoids, glucosinolates, benzoquinones, and cyanogenic compounds [9]. Essential oils and plant extracts have long been utilized as sources of bioactive molecules, particularly phenolic compounds and terpenes—two groups of allelochemicals recognized for their allelopathic potential. Generally, essential oils exhibit higher toxicity than extracts; however, if the extracts originate from plants known to produce toxic metabolites, these extracts may be more toxic than essential oils [10]. Although there is a lack of comparative studies on both types of extracts, published results indicate that essential oils have a stronger growth inhibitory effect on weeds than plant extracts [11]. Furthermore, essential oils demonstrate high biodegradability in the environment and relative safety for humans and other non-target organisms compared to synthetic pesticides [12]. The use of essential oils as biopesticides presents numerous challenges due to their inherent properties (lipophilicity and high volatility), production costs, and manufacturing limitations [3].

The Asteraceae family is a natural source of allelopathic sesquiterpenes and sesquiterpene lactones. They can inhibit the enzyme asparagine synthase, thereby preventing growth, and impair the respiration of mitochondrial cell organelles and the release of proteins into the plasma membrane [13]. In particular, the genus *Artemisia* L. comprises over 200 species worldwide, nine of which occur in the flora of Serbia: *A. vulgaris*, *A. absinthium*, *A. annua*, *A. pontica*, *A. petrosa*, *A. lobelii*, *A. maritima*, *A. campestris*, and *A. scoparia* [14]. Plants of this genus exhibit a broad spectrum of biological activities (antifungal, insecticidal, and herbicidal effects), including medicinal properties (anthelmintic, antimalarial, antispasmodic, anti-inflammatory, antirheumatic, and anticancer), due to the presence of different phytochemicals [15,16,17,18]. To our knowledge, there is no data in the literature regarding a dual approach to the allelopathic potential of plant extracts and essential oils from *A. absinthium* and *A. vulgaris* on seed germination and early seedling growth of weed species. Keeping all this in mind, the present study focused on (I) extraction of plant material from the aboveground parts of *A. absinthium* and *A. vulgaris* originating from Serbia; (II) analysis of total phenolic content (TPC) and evaluation of antioxidant activity of *A. absinthium* and *A. vulgaris* plant extracts (PEs); (III) identification and quantification of major phenolic compounds in the plant extracts (PEs) and terpenes in the essential oils (EOs) of *A. absinthium* and *A. vulgaris*; and (IV) *in vitro* evaluation of the allelopathic potential of the PEs and EOs of *A. absinthium* and *A. vulgaris* on seed germination and early seedling growth of two economically noxious weed species (*Amaranthus retroflexus* and *Setaria viridis*) widely distributed in arable fields.

## 2. Results

### 2.1. Chemical Analysis of A. absinthium and A. vulgaris Plant Extracts

The results of the UHPLC-DAD MS/MS analysis of the individual phenolic compounds in both *Artemisia* extracts are shown in Table 1. Chromatograms can be found in the Appendix A. In *A. absinthium*, two tested compounds, p-coumaric acid and luteolin, were below the detection limit. The most abundant phenolic compound was chlorogenic acid (1.694 ± 0.081 mg/g d.e.), while kaempferol-3-O-glucoside (0.197 ± 0.032 mg/g d.e.), rutin (0.135 ± 0.019 mg/g d.e.), isorhamnetin-3-O-rutinoside (0.090 ± 0.013 mg/g d.e.), and hyperoside (0.066 ± 0.011 mg/g d.e.) were among the other significant phenolic compounds likely responsible for the high antioxidant activity of the plant. In *A. vulgaris*, isorhamnetin was below the detection limit among the tested compounds. In this extract, chlorogenic acid (1.381 ± 0.075 mg/g d.e.) also contributed the most to the high phenolic content, followed by rutin (0.821 ± 0.046 mg/g d.e.), kaempferol-3-O-glucoside (0.579 ± 0.030 mg/g d.e.), and hyperoside (0.212 ± 0.015 mg/g d.e.).

The analysis of the total phenolic content shows that these compounds were present in significant amounts in both PEs, with the total phenolic content being higher in *A. vulgaris* (73.7 ± 2.5 mg GAE/g d.e.) than in *A. absinthium* (58.4 ± 2.4 mg GAE/g d.e.). The scavenging effect of the PEs of *A. absinthium* and *A. vulgaris* on DPPH radicals is shown in Figure 1. The PE of *A. vulgaris* with an IC_50_ value of 0.171 ± 0.01 mg/mL showed a greater reduction of DPPH and inhibited almost 90% of DPPH radicals at the highest concentration used (0.5 mg/mL), compared to *A. absinthium*, which had an IC_50_ value of 0.263 ± 0.01 mg/mL and inhibited approximately 75% of DPPH radicals at the same concentration. Both PEs showed moderate DPPH reduction compared to ascorbic acid, which had an IC_50_ value of 0.018 mg/mL. The results of the reducing antioxidant power of ferric (FRAP) analysis showed that *A. vulgaris* also had a higher capacity to reduce metal ions than *A. absinthium*. The FRAP value for the PE of *A. vulgaris* was 180.1 ± 21.4 µmol Fe^2+^/g d.e., while the PE of *A. absinthium* had a FRAP value of 138.0 ± 17.6 µmol Fe^2+^/g d.e. However, the reduction potential of both plant PEs was significantly lower than that of the standard used, as the FRAP value of ascorbic acid was 2576 ± 21 μmol Fe^2+^/g. Overall, the results showed that the PEs from *A. vulgaris* and *A. absinthium* demonstrate lower metal-reducing ability compared to their free radical scavenging ability.

### 2.2. Chemical Analysis of A. absinthium and A. vulgaris Essential Oils

The EOs of *A. absinthium* and *A. vulgaris* were extracted with yields of 0.088% (*v*/*w*) and 0.012% (*v*/*w*), respectively. The main oil components (compounds with contents higher than 1%) are detailed in Table 2. Chromatograms can be found in the Appendix A. The complete chemical analysis of the EOs from *A. absinthium* and *A. vulgaris*, which account for over 99% of the total oil mass, is presented in Appendix A. The EO of *A. absinthium* contained 19 compounds that exceeded 1% in quantity, representing 77.92% of the total oil mass. The most prominent chemical compound classes were monoterpenes (42.43%) and sesquiterpenes (28.58%), with mono- and sesquiterpene hydrocarbons contributing 7.90% and 18.96%, while oxygenated mono- and sesquiterpenes contributed 34.53% and 9.62%, respectively. The oil also contained geranyl-terpinene (1.51%) as a diterpene hydrocarbon, 13-epi-manool oxide (3.27%) as an oxygenated diterpene, and hexadecyl acetate (2.13%) as an ester compound. The most abundant components of the oil were trans-thujone (β-thujone) (18.9%), linalool (4.17%), and cis-ocimene epoxide (7.88%) as oxygenated monoterpenes; β-caryophyllene (6.0%), germacrene D (4.71%), and bicyclogermacrene (7.04%) as sesquiterpene hydrocarbons; and germacrene D-4-ol (5.35%) as an oxygenated sesquiterpene (alcohol).

In the EO of *A. vulgaris*, 21 components with contents of more than 1% were identified, accounting for 75.97% of the total oil mass. The most dominant components were gurjunene (10.41%), γ-humulene (6.67%), β-caryophyllene (5.81%), α-isocomene (5.15%), germacrene D (4.88%), and β-selinene (4.86%) as sesquiterpene hydrocarbons; cis-chrysanthenyl acetate (7.17%) as an oxygenated monoterpene; and davanone (5.62%) as an oxygenated sesquiterpene. Overall, the composition of the EO of *A. vulgaris* was dominated by sesquiterpene hydrocarbons (51.91%). The content of oxygenated sesquiterpenes and monoterpenes was 11.11% and 7.17%, respectively, with monoterpene hydrocarbons accounting for 3.93%, while geranyl benzoate, as a benzoate ester, accounted for 1.85% of the oil mass.

### 2.3. Assessment of the Allelopathic Potential of Plant Extracts of A. absinthium and A. vulgaris In Vitro

*A. absinthium* and *A. vulgaris* PEs at concentrations of 0.25%, 0.50%, 0.75%, 1.00%, 2.50%, and 5.00% significantly reduced the seed germination and early growth of seedlings of *A. retroflexus* and *S. viridis* (see Appendix A). The inhibition of seed germination of *A. retroflexus* at the highest *A. absinthium* PE concentrations (0.75%, 1.00%, 2.50%, and 5.00%) ranged from 78% to 100%, while the inhibition at the lowest *A. absinthium* PE concentrations (0.25% and 0.5%) was 8% to 43% (Figure 2a). Similarly, inhibition of all seedling growth parameters decreased in proportion to decreasing PE concentrations. The radicle length parameter was more sensitive, and an inhibition from 6% to 100% was recorded, while the inhibition for shoot length was lower (16–100%), except at the 0.25% PE concentration, where a stimulation of 13% was noted (Figure 2b). In general, the seeds of *S. viridis* were less sensitive than those of *A. retroflexus* to the different concentrations of PE *A. absinthium*. The inhibition of seed germination at the two highest concentrations (2.50% and 5.00%) was 92% and 100%, respectively. On the other hand, the concentrations of 1.00%, 0.75%, 0.50%, and 0.25% caused a negligible inhibition from 1% to 13% (Figure 2a). The inhibition increased in proportion to the increasing PE concentrations of *A. absinthium* for the seedling growth parameters (radicle and shoot length) and was similar for both parameters and ranged from 14% to 100% (Figure 2b,c). Based on the estimated regression parameters (Table 3), it was found that the concentrations of 0.54 ± 0.02% (*A. retroflexus*) and 1.51 ± 0.07% (*S. viridis*) of *A. absinthium* PE were sufficient to cause 50% inhibition of seed germination of the test plants. The EC_50_ values for seedling length showed a similar sensitivity for both weeds (Figure 2d). An exception was the radicle sensitivity (Figure 2c), where lower values were determined for *A. retroflexus* (0.18 ± 0.02%) than for *S. viridis* (0.55 ± 0.01%).

In general, the PE of *A. vulgaris* had a similar impact on seed germination and radicle length of *A. retroflexus* as *A. absinthium*. Conversely, the shoot length parameter was less sensitive, with stimulation of 19% and 35% recorded at concentrations of 0.50% and 0.25%, respectively (Figure 3b). Inhibition of seed germination of *S. viridis* was only noted at the highest concentrations (1.00%, 2.50%, and 5.00%) and ranged from 22% to 100% (Figure 3a). The parameters of seedling growth (radicle and shoot length) were more sensitive in both weed species in the treatment with PE of *A. vulgaris* than in the treatment with *A. absinthium*. The inhibition of radicle length ranged from 52% to 100% (Figure 3c), while shoot length ranged from 50% to 100% (Figure 3b). The calculated EC_50_ values of *A. vulgaris* PE for seed germination indicated slightly lower sensitivity for *A. retroflexus* (0.63 ± 0.01%) and *S. viridis* (1.74 ± 0.11%) in contrast to *A. absinthium* PE. The EC_50_ values for seed germination confirmed the higher sensitivity of *A. retroflexus* compared to *S. viridis* to both PEs. However, the EC_50_ values for seedling length revealed that both tested weeds were similarly sensitive to *A. absinthium* PE. On the other hand, *S. viridis* was more sensitive to PE of *A. vulgaris* than to *A. absinthium* (Figure 2d and Figure 3d).

### 2.4. Assessment of the Allelopathic Potential of Essential Oils of A. absinthium and A. vulgaris In Vitro

Increasing the EO concentrations of both *Artemisia* species enhanced the percentage of inhibition for all measured parameters in both weed species (Figure 4 and Figure 5). The absolute values of the measured parameters are shown in Appendix A. At the highest EO concentration of *A. absinthium* (0.50%), the seed germination of *A. retroflexus* was inhibited by 86%, while at lower concentrations (0.01–0.25%), the percentage of inhibition ranged from 1% to 48% (Figure 4a). All seedling growth parameters (shoot and radicle length) showed similar sensitivity, with inhibition of seedling length varying from 4% to 96% at all applied EO concentrations (Figure 4d). Inhibition of *S. viridis* germination at the highest concentrations of *A. absinthium* EO (0.10%, 0.25%, and 0.50%) ranged from 18% to 100%, with no significant differences observed between the lower concentrations (0.01%, 0.025%, and 0.05%) and the control. Although germination of *S. viridis* was inhibited by 18% at a concentration of 0.1%, seedling length was inhibited by 91% at the same EO concentration. The shoot length parameter also exhibited greater sensitivity, with inhibition ranging from 2% to 100% (Figure 4b). According to the estimated EC_50_ values for all measured parameters, *S. viridis* showed greater sensitivity than *A. retroflexus* (Table 4). Concentrations of 0.03 ± 0.00% and 0.28 ± 0.48% effectively achieved 50% inhibition of seedling length and seed germination of *S. viridis*, respectively.

The inhibition of seed germination of *A. retroflexus* after applying various concentrations of *A. vulgaris* EO ranged from 0% to 62% (Figure 5a). The inhibitory effect on seed germination was observed even at the lowest concentration (0.01%). Inhibition increased proportionally with rising *A. vulgaris* EO concentration for all seedling growth parameters (radicle and shoot length). Inhibition of shoot length ranged from 10% to 85% (Figure 5b), while radicle length inhibition varied from 6% to 82% (Figure 5c). *S. viridis* showed lower sensitivity to the EO of *A. vulgaris* than to *A. absinthium*. Significant differences were found only between the control and the highest concentrations (0.25% and 0.50%), at which germination of *S. viridis* was inhibited by 26% and 58%, respectively (Figure 5a). Higher percentages of inhibition were recorded for seedling length, which ranged from 13% to 96% (Figure 5d). Similar to the application of *A. absinthium* EO, the radicle length parameter was less sensitive than the shoot length. A stimulating effect (8%) was also observed at the lowest concentration (0.01%) of *A. vulgaris* EO (Figure 5c). Based on the calculated EC_50_ values for both EOs across all measured parameters, seedling growth (shoot and radicle length) demonstrated greater sensitivity than seed germination (Table 4). In contrast to *A. absinthium* EO, lower EC_50_ values were found for seed germination and seedling length of *A. retroflexus* (0.11 ± 0.04% and 0.02 ± 0.00%, respectively) compared to *S. viridis* (0.31 ± 0.07% and 0.05 ± 0.00%, respectively) after the application of *A. vulgaris* EO.

## 3. Discussion

Due to their chemical diversity and biological activity, interest in the biological properties of *Artemisia* species has increased. The results demonstrate that the PEs and EOs of the two examined *Artemisia* species are rich in phenols and volatiles and possess allelopathic properties.

The findings revealed that the PE of *A. vulgaris* contained higher levels of phenols (73.7 ± 2.5 mg GAE/g d.e.) than *A. absinthium* (58.4 ± 2.4 mg GAE/g d.e.). Additionally, the extract of *A. vulgaris* was found to be effective in scavenging the DPPH radical and had a greater ability to reduce metal ions, suggesting superior antioxidant activity compared to *A. absinthium*. The high antioxidant potential of *A. vulgaris* extracts, with chlorogenic acid derivatives and flavonoids as main constituents, has also been reported [20].

Despite the higher content of phenols and antioxidant activity of *A. vulgaris* PE, the PE of *A. absinthium* showed a greater inhibitory effect on seed germination of both weed species. Furthermore, *A. absinthium* PE (EC_50_ = 0.57 ± 0.03%) proved to be more effective than *A. vulgaris* PE (EC_50_ = 0.82 ± 0.02%) in inhibiting seedling growth (radicle and shoot length) of *A. retroflexus*. An exception was *S. viridis*, where the EC_50_ values of *A. vulgaris* PE for shoot and radicle length (0.41 ± 0.04% and 0.23 ± 0.01%, respectively) were lower than those of *A. absinthium* (0.64 ± 0.15% and 0.55 ± 0.01%, respectively). A similar phenomenon was described by Marcinkeviciene et al. [21], where the aqueous extract of *A. vulgaris* leaves, which had a higher content of phenolics, showed weaker inhibition of seed germination, root, and shoot growth of winter wheat and oilseed rape than the aqueous extract of *A. vulgaris* roots with a lower content of phenolic compounds. Radicle length was the most sensitive parameter following the application of both PEs. Lower EC_50_ values for radicle compared to shoot length were also reported after applying the aqueous leaf extract of *A. absinthium* on *Parthenium hysterophorus* [18]. The shoot growth of *A. retroflexus* was stimulated after applying the lowest concentration (0.25%) of both PEs, and a relatively significant stimulation (19%) was also observed after the application of 0.50% of *A. vulgaris* PE. The stimulatory effects following the application of lower concentrations of various PEs are already well documented and have been observed in *A. vulgaris* [22,23]. In both PEs, chlorogenic acid was the main constituent, but a higher concentration of this phenolic acid was found in the PE of *A. absinthium* (1.694 ± 0.081 mg/g) than in the PE of *A. vulgaris* (1.381 ± 0.075 mg/g), which could explain the greater allelopathic effect of the PE of *A. absinthium*. The phytotoxic effect of chlorogenic acid and its impact on seed germination, root, and shoot length of various plants have already been reported [24]. In addition, the allelopathic potential of various solvent extracts of *Artemisia* species with chlorogenic acid as one of the dominant compounds has already been confirmed [25,26]. Xu et al. [27] discovered DHAR1 (dehydroascorbate reductase 1), an enzyme responsible for ascorbate regeneration in plants, as the target site of chlorogenic acid, whose inhibition of activity led to the accumulation of H_2_O_2_ and the reduction of proteins involved in water transport and photosynthesis. Chlorogenic acid was also identified as a dominant compound in other *Artemisia* species (*A. alba*, *A. annua*, *A. campestris*, *A. ponica*, and *A. vulgaris*) growing in Serbia [28].

The EO of *A. absinthium* consisted of 34.53% of components containing more than 1% belonging to oxygenated monoterpenes, while sesquiterpene hydrocarbon compounds, accounting for 51.91%, were the major constituents of the EO of *A. vulgaris*. Previous studies have demonstrated that monoterpenes, particularly oxygenated monoterpenes, can have toxic effects on the growth of numerous plants [29]. A comparison of the composition of *A. vulgaris* EO with that of the same species from Serbia shows relatively good agreement between the results [30,31,32], although certain differences in the composition and concentration of individual components can be explained by the various geo-eco-climatic conditions under which the plants were grown. Among other findings, these differences were corroborated by Ickovski et al. [31], who analyzed EOs extracted from *A. vulgaris* across 12 different localities. As in the case of the EO of *A. vulgaris*, certain differences in the composition and content of the individual components of the EO of *A. absinthium* can be explained by the different geo-eco-climatic conditions under which the plants were grown, which was confirmed by Ickovski et al. [31], who analyzed EOs extracted from 12 different locations, and Blagojević et al. [32], who analyzed 3 EOs isolated from plants grown in different localities. In general, oxygenated monoterpenes were the most dominant components in all analyzed EOs (from 72.3% to 89.1% by Blagojević et al. [32] and from 37.1% to 73.8% by Ickovski et al. [31]), with β-thujone being the most dominant component in most of them, which is consistent with our results. Additionally, the high content of β-thujone (25.75%) in the EOs isolated from the leaves of *A. absinthium* originating from Tunisia has already been reported [33]. Similarly, Fouad et al. [34] found that β-thujone (35.6%) was the main constituent of the EO of *A. absinthium* originating from Morocco.

Most studies investigating the allelopathic potential of *Artemisia* species have focused on examining the herbicidal properties of their EOs, with only a few exploring aqueous or alcoholic extracts [35]. The complex chemical composition of essential oils makes isolating their components and determining their potential mode of action difficult. The phytotoxic effects of essential oils have been associated with plant growth reduction, leaf chlorosis, changes in plant cells, mitosis, cellular respiration, chlorophyll content, membrane depolarization, ion leakage, cuticular waxes, oxidative stress, and microtubular polymerization [36]. This study confirms that the EOs of both *Artemisia* species exhibit a significantly higher inhibitory effect on the tested economically noxious weed species compared to the PEs, suggesting a greater allelopathic potential of the terpenes than the phenolic compounds. The calculated EC_50_ values for both EO and PE indicate that seedling growth is a more sensitive parameter than seed germination for both weed species. Based on the estimated regression parameters, the EC_50_ values for the investigated EOs ranged from 0.11 ± 0.04% to 0.37 ± 0.15% for seed germination and from 0.02 ± 0.00% to 0.06 ± 0.00% for seedling length. Other authors have also pointed out a greater sensitivity of seedling growth compared to seed germination [37,38]. The EC_50_ values determined for shoot and radicle length (0.02 ± 0.00% and 0.03 ± 0.00%) of *A. retroflexus* treated with *A. vulgaris* EO were slightly lower than the values reported by Han et al. [39] regarding the effects of this EO on the same test plant and parameters (EC_50_–0.356 mg/mL for shoot length and 0.308 mg/mL for root length). The lowest EC_50_ values for *A. absinthium* EO in terms of seed germination and seedling length were estimated for *S. viridis* (0.28 ± 0.48% and 0.03 ± 0.00%, respectively). The inhibitory effect of *A. absinthium* EO has already been confirmed in *A. retroflexus*, *Poa annua* [40], and *Sinapis arvensis* [34].

## 4. Materials and Methods

### 4.1. Collection and Extraction of Plant Material

Aboveground plant parts of *A. absinthium* and *A. vulgaris* were collected during the flowering period in July and August 2022 in Vratna (44°22′57″ N; 22°20′31″ E) and Rumenka (45°18′28″ N; 19°43′40″ E), respectively. The collected plant material was dried in the shade at a temperature of 22 ± 1 °C for three weeks and then stored in paper bags in a dry place until extraction. The seeds of *A. retroflexus* were collected in the field around Noćaj (44°55′33.5″ N 19°32′43.6″ E), and seeds of *S. viridis* were collected in the field around Majur (44°46′10.1″ N 19°40′30.0″ E) on non-arable habitat in October 2022.

The EOs were obtained from the dried aboveground plant material of *A. absinthium* (40 kg) and *A. vulgaris* (61 kg) by steam distillation in a small-scale distillation unit at the Institute of Field and Vegetable Crops in Novi Sad [41]. After drying over anhydrous sodium sulfate (Na_2_SO_4_), the EOs were stored in an amber bottle in a cold and dark place until use.

The PEs of *A. absinthium* and *A. vulgaris* were obtained by successive solvent extraction using seven solvents (hexane, ethyl acetate, acetone, acetonitrile, ethanol, methanol, and distilled water) to extract phenolic compounds with a broad polarity spectrum. The solvents were used successively in a serial manner in a 1:4 *w*/*v* ratio and sonicated for 15 min at 40 °C (frequency 70 kHz and power 240 W). After that, every aliquot of the extract was filtered separately through filter paper and evaporated to dryness at 40 °C using a vacuum rotary evaporator. After evaporation, the dry residue of each plant extract was merged and subjected to a lyophilization process. The PEs from both *Artemisia* species were stored dry until use.

### 4.2. Chemical Analysis of A. absinthium and A. vulgaris Plant Extracts

Quantitative and qualitative analyses of phenolic compounds in PEs were performed using a Dionex Ultimate 3000 UHPLC system equipped with a diode array detector (DAD) connected to a TSQ Quantum Access Max triple-quadrupole mass spectrometer (ThermoFisher Scientific, Basel, Switzerland), as previously described by Gašić et al. [42]. To prepare the extracts for analysis, 5 mg of plant extract was weighed and mixed with 96% methanol (*w*/*v* = 1:10) in an ultrasonic bath for 30 min. After centrifugation at 10,000× *g* for 10 min, the supernatants were filtered through 0.2 µm cellulose filters (Agilent Technologies, Santa Clara, CA, USA) and stored at 4 °C until analysis. The results were expressed as mg/g of dry extract.

The total phenolic content (TPC) of the PEs of *A. absinthium* and *A. vulgaris* was determined using the modified Folin–Ciocalteu method [43]. The TPC results were reported as mg gallic acid per g of dry extract (mg GAE/g d.e.). The antioxidant activity of the PEs was evaluated by determining the free radical scavenging activity using the modified DPPH method [44] and measuring the reducing potential using the modified FRAP assay [45]. The IC_50_ value (the concentration of the sample required to scavenge 50% of DPPH) was calculated, and the reducing potential results were expressed as mmol of ferric ions (Fe^2+^) per g of dry extract. Ascorbic acid served as a positive control for both methods.

### 4.3. Chemical Analysis of A. absinthium and A. vulgaris Essential Oils

Identification of the compounds in the EOs of *A. vulgaris* and *A. absinthium* was carried out using gas chromatography–mass spectrometry (GC-MS, model CP-3800/Saturn 2200), which was equipped with a split/splitless injector and a DB-5MS column (30 m × 0.25 mm and 0.25 µm film thickness). The Wiley 7.0 mass spectral library was utilized, and the obtained experimental retention indices (RIs) were compared with literature data [19]. Quantitative analysis of the EOs was performed on a gas chromatograph (GC, Agilent 7890A) with a split/splitless injector, a DB-5MS column (30 m × 0.25 mm, 0.25 µm film thickness), and a flame ionization detector (FID). In both cases, 1 µL of the hexane solution of the EO sample (1% solution) was injected in split mode (1:20). For the GC-MS analysis, the injector temperature was set to 250 °C, while the ion trap and transfer line temperatures were set to 250 °C and 280 °C, respectively. Helium was used as the carrier gas with a flow rate of 1 mL/min. The column temperature was programmed to increase linearly from 50 °C to 250 °C at a rate of 4 °C/min, with an isothermal hold at 250 °C for 10 min. The mass detector was operating in electron impact (EI) mode at 70 eV, with the mass range of 40–600 *m*/*z*. Total ion current (TIC) was used to record the chromatograms obtained. In the GC-FID analysis, the injector and detector temperatures were set to 250 °C and 300 °C, respectively. Hydrogen was used as the carrier gas at a flow rate of 1 mL/min, and the temperature program of the column was identical to that used in the GC-MS analysis.

### 4.4. Assessment of the Allelopathic Potential of Plant Extracts and Essential Oils of A. absinthium and A. vulgaris In Vitro

Three factors were considered in the *in vitro* evaluation of allelopathic potential: (I) weed seeds (*A. retroflexus* and *S. viridis*); (II) PEs of *A. absinthium* and *A. vulgaris* in the concentrations 0.25%, 0.50%, 0.75%, 1.00%, 2.50%, and 5.00% *w*/*v* (prepared in distilled water); and (III) EOs of *A. absinthium* and *A. vulgaris* in the concentrations 0.01, 0.025, 0.05, 0.10, 0.25, and 0.50% *v*/*v* (prepared in distilled water with the surfactant Tween 20, conc. 0.05% *v*/*v*). Distilled water and distilled water with added surfactant were used as controls. The test seeds were soaked in a 2% thiourea solution [SC(NH_2_)_2_] for 24 h to overcome their dormancy. The seeds were then surface sterilized in 3% hydrogen peroxide (H_2_O_2_) for 5 min and then rinsed with distilled water. Twenty-five soaked seeds were placed in a Petri dish (90 mm diameter) with filter paper, and 5 mL of the PE or EO solution was added. All Petri dishes were sealed with Parafilm to avoid evaporation and placed in an incubator (Memmert, Schwabach, Germany) at 27 ± 1 °C in the dark conditions. All treatments were carried out in four repetitions, and the experiment was repeated twice. The percentage of germinated seeds was calculated, and early seedling growth (radicle length, shoot length, and their sum, expressed as seedling length) was measured after seven days.

### 4.5. Statistical Analysis

The data were analyzed by one-way analysis of variance (ANOVA) using the software package STATISTICA 8.0. When F values were statistically significant (*p* < 0.05), treatments were compared using Fisher’s least significant difference (LSD) test. Data on inhibition of germination and early seedling growth (seedling length, shoot, and radicle length) were analyzed with the program R using the statistical add-on package “drc” with a four-parameter log-logistic nonlinear regression model [46]: Y = C + (D − C)/1 + exp (B (logX − logE))(1)
where Y is the response (e.g., percent of inhibition), C is the lower limit, D is the upper limit, E is the dose resulting in a 50% response between the upper and lower limits (also known as inflection point I_50_ or EC_50_), the parameter B denotes the relative slope around E, and X is the concentration of PEs and EOs of *A. absinthium* and *A. vulgaris*.

## 5. Conclusions

The analysis of phenolic compounds from both *Artemisia* species growing in Serbia in this study demonstrates that *A. absinthium* and *A. vulgaris* PEs are rich sources of phenolic and flavonoid compounds with pronounced antioxidant activity. Additionally, this study reveals that *A. absinthium* and *A. vulgaris* EOs are rich sources of terpenes with high allelopathic potential. The EOs of both *Artemisia* species have a greater inhibitory effect on the germination and early seedling growth of *A. retroflexus* and *S. viridis* compared to PEs. The relevant data lead to the conclusion that the EOs of *A. absinthium* and *A. vulgaris* could serve as potential sources of natural-based weed control molecules. The prospects for practical application will be investigated through further studies, which will include the identification and isolation of the most effective allelochemicals from this source and validation of the present results in vivo and under field conditions, as well as regarding different weeds and crops.

## Figures and Tables

**Figure 1 plants-14-01663-f001:**
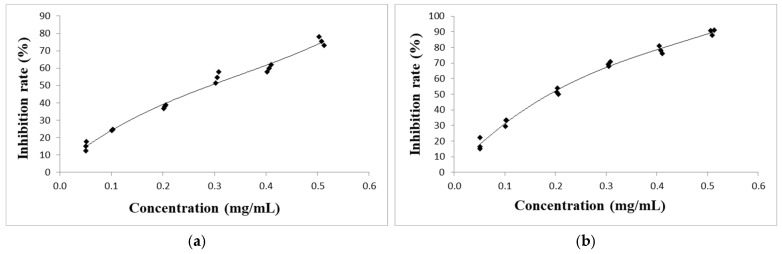
The scavenging effect of PEs of *A. absinthium* (**a**) and *A. vulgaris* (**b**) on DPPH radical.

**Figure 2 plants-14-01663-f002:**
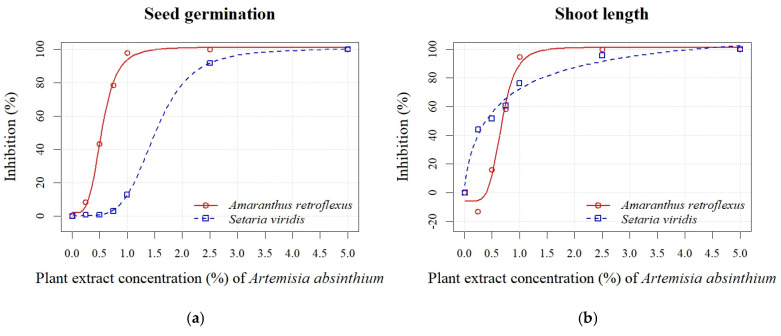
Effects of different PE concentrations (0.25%, 0.50%, 0.75%, 1.00%, 2.50%, and 5.00%) of *A. absinthium* on seed germination (**a**), shoot length (**b**), radicle length (**c**), and seedling length (**d**) of *A. retroflexus* and *S. viridis*.

**Figure 3 plants-14-01663-f003:**
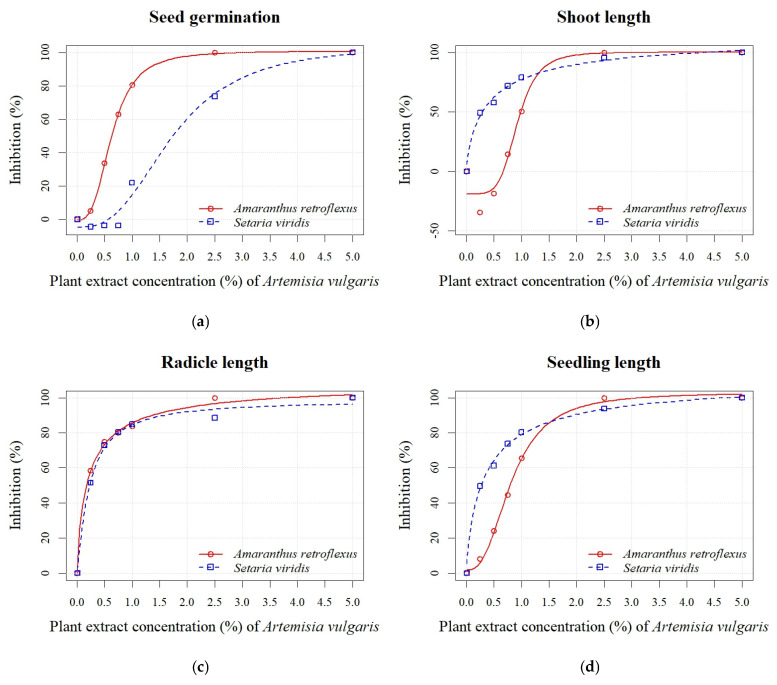
Effects of different PE concentrations (0.25%, 0.50%, 0.75%, 1.00%, 2.50%, and 5.00%) of *A. vulgaris* on seed germination (**a**), shoot length (**b**), radicle length (**c**), and seedling length (**d**) of *A. retroflexus* and *S. viridis*.

**Figure 4 plants-14-01663-f004:**
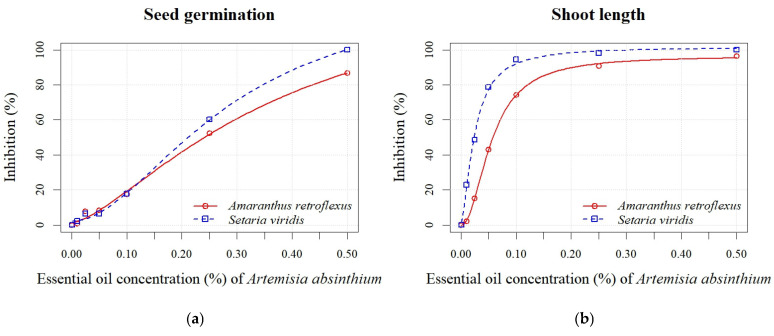
Effects of different EO concentrations (0.01, 0.025, 0.05, 0.10, 0.25, and 0.50%) of *A. absinthium* on seed germination (**a**), shoot length (**b**), radicle length (**c**), and seedling length (**d**) of *A. retroflexus* and *S. viridis*.

**Figure 5 plants-14-01663-f005:**
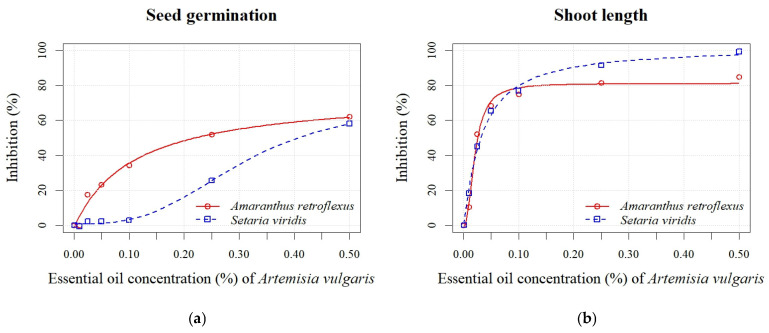
Effects of different EO concentrations (0.01, 0.025, 0.05, 0.10, 0.25, and 0.50%) of *A. vulgaris* on seed germination (**a**), shoot length (**b**), radicle length (**c**), and seedling length (**d**) of *A. retroflexus* and *S. viridis*.

**Table 1 plants-14-01663-t001:** Quantification of identified phenols in plant extracts (PEs) of *A. absinthium* and *A. vulgaris*.

Chemical Class	Component	RT (min)	Molecular Ions [M-H]^−^	Q1 (*m*/*z*)	Q3 (*m*/*z*)	Content (mg/g)	
						*A. absinthium*	*A. vulgaris*
Flavonoid	Epicatechin	5.34	289.071	245.06	203.05	0.027 ± 0.001	0.074 ± 0.004
	Rutin	6.41	609.146	301.04	179.00	0.135 ± 0.019	0.821 ± 0.046
	Hyperoside	6.41	463.088	301.04	179.00	0.066 ± 0.011	0.212 ± 0.015
	Quercetin	8.15	301.035	179.00	151.00	0.017 ± 0.003	0.013 ± 0.001
	Luteolin	8.17	285.040	241.03	227.02	n.d. *	0.053 ± 0.005
	Kaempferol	8.92	285.041	255.03	211.01	0.004 ± 0.000	0.030 ± 0.009
	Isorhamnetin	9.20	315.051	300.05	272.03	0.039 ± 0.002	n.d.
Flavonoid-3-o-glycosides	Isorhametin-3-O-rutinoside	6.67	623.162	315.05	300.04	0.090 ± 0.013	0.053 ± 0.003
	Isorhametin-3-O-glucoside	6.90	477.104	314.05	299.05	0.023 ± 0.003	0.015 ± 0.001
	Kaempferol-3-O-glucoside	6.93	447.094	285.04		0.197 ± 0.032	0.579 ± 0.030
Hydroxycinnamic acids	Chlorogenic acid	4.33	353.089	191.03		1.694 ± 0.081	1.381 ± 0.075
	Caffeic acid	5.81	179.035	135.02		0.058 ± 0.002	0.037 ± 0.003
	p-Coumaric acid	6.58	163.040	119.03		n.d.	0.006 ± 0.001
Coumarin	Aesculetin	5.83	177.019	133.01		0.034 ± 0.000	0.025 ± 0.001

Chemical classes, target compounds, retention times, molecular ions [M-H]^−^, MS^2^ fragments (Q1 and Q3), and content of compounds are presented. All presented compounds were confirmed using available standards. * n.d.—not detected; values of content are presented as mean ± standard deviation.

**Table 2 plants-14-01663-t002:** The main components of *A. absinthium* and *A. vulgaris* essential oils (EOs).

Component	^a^ RI_EXP_	^b^ RI_LIT_	Content (%)	
			*A. absinthium*	*A. vulgaris*
Sabinene	969	969	3.94 ± 0.11	1.14 ± 0.05
Myrcene	988	988	1.77 ± 0.05	2.79 ± 0.07
p-Cymene	1020	1020	2.19 ± 0.06	n.d. *
Linalool	1094	1095	4.17 ± 0.17	n.d.
Trans-thujone	1112	1112	18.90 ± 0.86	n.d.
Cis-epoxy ocimene	1127	1128	7.88 ± 0.42	n.d.
4-Terpineol	1173	1174	1.03 ± 0.09	n.d.
Cumin aldehyde	1238	1238	1.52 ± 0.06	n.d.
Cis-chrysanthenyl acetate	1262	1261	n.d.	7.17 ± 0.16
Bornyl acetate	1285	1284	1.03 ± 0.08	n.d.
α-Cubebene	1346	1345	n.d.	1.08 ± 0.05
Cyclosativene	1370	1369	n.d.	1.48 ± 0.09
Modheph-2-ene	1381	1382	n.d.	1.09 ± 0.06
α-Isocomene	1386	1387	n.d.	5.15 ± 0.22
Cyperene	1397	1398	n.d.	2.54 ± 0.09
β-Caryophyllene	1416	1417	6.00 ± 0.47	5.81 ± 0.17
α-Humulene	1451	1452	n.d.	3.43 ± 0.08
4,5-Di-epi-aristolochene	1471	1471	n.d.	1.42 ± 0.06
γ-Gurjunene	1476	1475	1.21 ± 0.05	10.41 ± 0.31
γ-Humulene	1480	1481	n.d.	6.67 ± 0.19
Germacrene D	1483	1484	4.71 ± 0.21	4.88 ± 0.11
β-Selinene	1491	1491	n.d.	4.86 ± 0.07
Bicyclogermacrene	1500	1500	7.04 ± 0.47	1.15 ± 0.03
δ-Cadinene	1522	1522	n.d.	1.94 ± 0.05
Germacrene D-4-ol	1573	1574	5.35 ± 0.39	n.d.
Caryophyllene oxide	1581	1582	n.d.	3.75 ± 0.14
Neryl 2-methyl-butanoate	1584	1584	3.23 ± 0.11	n.d.
Davanone	1587	1587	1.04 ± 0.07	5.62 ± 0.20
β-Himachalene oxide	1615	1615	n.d.	1.74 ± 0.06
Geranyl benzoate	1957	1958	n.d.	1.85 ± 0.04
Geranyl-α-terpinene	1961	1962 **	1.51 ± 0.06	n.d.
Hexadecyl acetate	2004	2003	2.13 ± 0.11	n.d.
13-Epi-manool oxide	2010	2009	3.27 ± 0.13	n.d.
Total identified (%)			77.92	75.97
Monoterpene hydrocarbons			7.90	3.93
Oxygenated monoterpenes			34.53	7.17
Sesquiterpene hydrocarbons			18.96	51.91
Oxygenated sesquiterpenes			9.62	11.11
Diterpene hydrocarbons			1.51	n.d.
Oxygenated diterpenes			3.27	n.d.
Other			2.13	1.85

^a^ RI_EXP_—retention indexes experimentally determined (calculated relative to C6-C28 n-alkanes on the DB-5 column); ^b^ RI_LIT_—retention indexes from literature data [19]; * n.d.—not detected; ** retention indexes from the NIST (National Institute of Standards and Technology, U.S. Department of Commerce) database.

**Table 3 plants-14-01663-t003:** Regression parameters for seed germination, shoot length, radicle length, and seedling length for the test plants (*A. retroflexus* and *S. viridis*) after application of PEs of *A. absinthium* and *A. vulgaris*.

Plant Extract of *A. absinthium*				
Measured parameters	Regression parameters			
	B	C	D	EC_50_ (I_50_)
*A. retroflexus*				
Seed germination	−4.07 ± 0.49	2.01 ± 2.48	101.25 ± 2.02	0.54 ± 0.02
Shoot length	−5.26 ± 0.48	−5.97 ± 2.16	101.33 ± 1.98	0.67 ± 0.02
Radicle length	−0.93 ± 0.15	0.02 ± 2.31	107.36 ± 3.85	0.18 ± 0.02
Seedling length	−2.94 ± 0.47	5.66 ± 3.28	102.55 ± 2.15	0.57 ± 0.03
*S. viridis*				
Seed germination	−4.74 ± 0.49	0.18 ± 1.24	100.27 ± 2.37	1.51 ± 0.07
Shoot length	−0.78 ± 0.12	0.27 ± 2.34	123.33 ± 9.88	0.64 ± 0.15
Radicle length	−2.09 ± 0.11	−0.59 ± 1.34	98.97 ± 1.28	0.55 ± 0.01
Seedling length	−1.02 ± 0.09	−0.08 ± 1.73	112.57 ± 3.95	0.56 ± 0.05
Plant extract of *A. vulgaris*				
Measured parameters	Regression parameters			
	B	C	D	EC_50_ (I_50_)
*A. retroflexus*				
Seed germination	−3.02 ± 0.16	−0.36 ± 1.28	100.79 ± 1.15	0.63 ± 0.01
Shoot length	−5.06 ± 0.78	−18.86 ± 3.06	100.22 ± 3.48	0.93 ± 0.03
Radicle length	−0.83 ± 0.10	−0.01 ± 1.41	108.96 ± 3.28	0.21 ± 0.02
Seedling length	−2.63 ± 0.19	1.81 ± 1.44	102.87 ± 1.50	0.82 ± 0.02
*S. viridis*				
Seed germination	−2.78 ± 0.26	−4.76 ± 1.72	104.63 ± 4.46	1.74 ± 0.11
Shoot length	−0.79 ± 0.08	0.14 ± 1.38	115.48 ± 4.53	0.41 ± 0.04
Radicle length	−1.21 ± 0.12	−0.10 ± 1.46	98.62 ± 1.97	0.23 ± 0.01
Seedling length	−0.83 ± 0.06	0.05 ± 1.08	111.37 ± 3.03	0.34 ± 0.02

Regression parameters (means ± standard deviation) showing the slope (B), lower limit (C), upper limit (D), and 50% reduction (EC_50_).

**Table 4 plants-14-01663-t004:** Regression parameters for seed germination, shoot length, radicle length, and seedling length for the test plants (*A. retroflexus* and *S. viridis*) after application of EOs of *A. absinthium* and *A. vulgaris*.

Essential Oil of *A. absinthium*				
Measured parameters	Regression parameters			
	B	C	D	EC_50_ (I_50_)
*A. retroflexus*				
Seed germination	−1.47 ± 0.31	1.33 ± 1.79	142.65 ± 38.80	0.37 ± 0.15
Shoot length	−2.05 ± 0.14	−0.37 ± 1.32	96.39 ± 1.56	0.06 ± 0.00
Radicle length	−1.29 ± 0.09	−1.31 ± 1.58	99.76 ± 2.67	0.05 ± 0.00
Seedling length	−1.65 ± 0.09	−0.71 ± 1.12	97.46 ± 1.48	0.06 ± 0.00
*S. viridis*				
Seed germination	−1.88 ± 0.31	2.03 ± 1.62	133.88 ± 17.92	0.28 ± 0.48
Shoot length	−1.61 ± 0.08	0.99 ± 1.31	101.67 ± 1.04	0.02 ± 0.00
Radicle length	−2.92 ± 0.26	0.28 ± 1.37	96.03 ± 1.40	0.04 ± 0.00
Seedling length	−1.75 ± 0.10	1.81 ± 1.37	100.95 ± 1.12	0.03 ± 0.00
Essential oil of *A. vulgaris*				
Measured parameters	Regression parameters			
	B	C	D	EC_50_ (I_50_)
*A. retroflexus*				
Seed germination	−0.99 ± 0.19	−1.61 ± 2.51	75.79 ± 10.74	0.11 ± 0.04
Shoot length	−2.20 ± 0.17	1.32 ± 1.51	81.08 ± 1.09	0.02 ± 0.00
Radicle length	−1.67 ± 0.13	−1.95 ± 1.58	79.27 ± 1.53	0.03 ± 0.00
Seedling length	−1.91 ± 0.15	−1.62 ± 1.50	80.03 ± 1.24	0.02 ± 0.00
*S. viridis*				
Seed germination	−2.95 ± 1.29	0.72 ± 1.09	72.09 ± 18.44	0.31 ± 0.07
Shoot length	−1.15 ± 0.06	−0.73 ± 1.20	101.44 ± 1.71	0.03 ± 0.00
Radicle length	−1.72 ± 0.14	−3.99 ± 1.26	102.49 ± 4.20	0.14 ± 0.01
Seedling length	−1.04 ± 0.06	−0.86 ± 1.18	104.71 ± 2.54	0.05 ± 0.00

Regression parameters (means ± standard deviation) showing the slope (B), lower limit (C), upper limit (D), and 50% reduction (EC_50_).

## Data Availability

The original contributions presented in this study are included in the article/Appendix A. Further inquiries can be directed to the corresponding author(s).

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
