# Peer review of "Allelopathic Potential of Artemisia absinthium and Artemisia vulgaris from Serbia: Chemical Composition and Bioactivity on Weeds"

_plants, 2025, doi:10.3390/plants14111663_

Round 1
Reviewer 1 Report
Comments and Suggestions for Authors
The manuscript presents a well-structured and comprehensive study on the allelopathic potential of plant extracts and essential oils from Artemisia absinthium and Artemisia vulgaris against two important weed species. The methodology is sound, data are clearly presented, and the findings are relevant to sustainable weed management strategies. The work fits well within the aims and scope of Plants, particularly under the themes of plant biochemistry, allelopathy, and agroecology.
However, some aspects of the manuscript need improvement or clarification before it can be recommended for publication.
While the allelopathic potential of Artemisia species has been previously studied, this work offers added value through comparative analysis of extracts and essential oils, detailed EC₅₀ calculations, and a solid chemical profile. However, the Introduction should more clearly highlight what differentiates this study from previous work, e.g., regional specificity, dual approach (PE and EO), or novelty in target species.
The finding that A. vulgaris had higher phenolic content and antioxidant capacity yet showed weaker allelopathic effects than A. absinthium is interesting. Are specific compounds (e.g., β-thujone) more phytotoxic than phenols? Was compound bioavailability considered?
A brief biological interpretation of each parameter would benefit the description of the nonlinear regression model (lines 419–426). Also, clarify whether replicates were biological or technical. Figures 2–5 and Tables are numerous and detailed but require better integration into the main text. Ensure all figures are explicitly referred to when results are discussed. Supplementary tables are well-structured but could be better referenced (e.g., “see Table S2” rather than “Supplementary Table S2”).
Use consistent terminology: e.g., “radical” should be corrected to “radicle” throughout the manuscript.
Clarify what “seedling length” refers to (sum of shoot + radicle?) early in Methods or Figure legends.
Comments on the Quality of English LanguageWhile generally readable, the manuscript would benefit from English editing to improve flow, reduce redundancy, and correct minor grammatical issues (see minor comments below).
Title: Consider simplifying to “Allelopathic Potential of Artemisia absinthium and A. vulgaris from Serbia: Chemical Composition and Bioactivity on Weeds” to enhance readability.
Lines 34–78: Add a brief paragraph outlining previous findings on EO vs. PE activity to justify the study design.
Lines 229–233: Rephrase for clarity. Suggest: “The regression parameters (B, C, D, and EC₅₀) indicated that seedling growth was more sensitive than seed germination in both species.”
Figures: Make sure all figures have fully descriptive captions. Current ones are too brief (e.g., Fig. 3).
Lines 421–426: There is a bracket mismatch in the equation. Ensure the mathematical expression is correctly formatted.
Reference Style: Ensure journal titles are consistently abbreviated or spelled out as per Plants guidelines.
Author Response
Dear Reviewer 1,
Thank Reviewer 1 for the valuable corrections and all the suggestions about our paper. We have done our best to improve it following the remarks we received. All comments to amend the manuscript were accepted, and the manuscript was changed following suggestions. We did our best to clarify parts of the manuscript that needed clarification. Changes that we have made according to your suggestions are made throughout the manuscript and highlighted in a yellow background. Below are our point-to-point responses to the general as well as each specific comment and suggestion. In addition, English language editing and style polishing were done.
Reviewer 1
The manuscript presents a well-structured and comprehensive study on the allelopathic potential of plant extracts and essential oils from Artemisia absinthium and Artemisia vulgaris against two important weed species. The methodology is sound, data are clearly presented, and the findings are relevant to sustainable weed management strategies. The work fits well within the aims and scope of Plants, particularly under the themes of plant biochemistry, allelopathy, and agroecology.
However, some aspects of the manuscript need improvement or clarification before it can be recommended for publication.
- While the allelopathic potential of Artemisia species has been previously studied, this work offers added value through comparative analysis of extracts and essential oils, detailed EC₅₀ calculations, and a solid chemical profile. However, the Introduction should more clearly highlight what differentiates this study from previous work, e.g., regional specificity, dual approach (PE and EO), or novelty in target species.
As suggested, we highlight the differences between our study and previous.
According to our knowledge, there are no data in the literature with a dual approach to the allelopathic potential of plant extracts and essential oils from A. absinthium and A. vulgaris on seed germination and early seedling growth of weed species.
-The finding that A. vulgaris had higher phenolic content and antioxidant capacity yet showed weaker allelopathic effects than A. absinthium is interesting. Are specific compounds (e.g., β-thujone) more phytotoxic than phenols? Was compound bioavailability considered.
Thank you for your suggestion. We added more information about essential oils and phenolic compounds in the Introduction (lines 56-68).
In general, essential oils have a higher toxicity than extracts, but if the extracts are obtained from plants known to produce toxic metabolites, the extracts may be more toxic than essential oils (Ferraz et al., 2022). The use of essential oils as biopesticides is associated with numerous challenges due to their inherent properties (lipophilicity and high volatility), production costs, and manufacturing limitations (Gupta et al., 2023).
Ferraz, C. A; Pastorinho, M. R.; Palmeira-de-Oliveira, A.; Sousa, A.C.A. Ecotoxicity of plant extracts and essential oils: A review. Environ. Pollut. 2022, 292, 118319. https://doi.org/10.1016/j.envpol.2021.118319
Gupta, I.; Singh, R.; Muthusamy, S.; Sharma, M.; Grewal, K.; Singh, H.P.; Batish, D.R. Plant Essential Oils as Biopesticides: Applications, Mechanisms, Innovations, and Constraints. Plants, 2023, 12, 2916. https://doi.org/10.3390/plants12162916
- A brief biological interpretation of each parameter would benefit the description of the nonlinear regression model (lines 419–426). Also, clarify whether replicates were biological or technical. Figures 2–5 and Tables are numerous and detailed but require better integration into the main text. Ensure all figures are explicitly referred to when results are discussed. Supplementary tables are well-structured but could be better referenced (e.g., “see Table S2” rather than “Supplementary Table S2”).
As you suggested, we have checked the references to all the Figures and Tables in the text. Also, we better refer to Table S2.
-Use consistent terminology: e.g., “radical” should be corrected to “radicle” throughout the manuscript.
As you suggested, we corrected the term “radical” with “radicle”.
-Clarify what “seedling length” refers to (sum of shoot + radicle?) early in Methods or Figure legends.
We appreciate your input. We clarified the term “seedling length” in M&M.
The percentage of germinated seeds was calculated, and early seedling growth (radicle length, shoot length, and their sum, expressed as seedling length) was measured after seven days.
Comments on the Quality of English Language
While generally readable, the manuscript would benefit from English editing to improve flow, reduce redundancy, and correct minor grammatical issues (see minor comments below).
-Title: Consider simplifying to “Allelopathic Potential of Artemisia absinthium and A. vulgaris from Serbia: Chemical Composition and Bioactivity on Weeds” to enhance readability.
We agree with your proposal. We have changed the title of the manuscript.
-Lines 34–78: Add a brief paragraph outlining previous findings on EO vs. PE activity to justify the study design.
As the reviewer noted, we added a brief paragraph outlining previous findings on EOs and PEs.
Essential oils and plant extracts have long been used as a source of bioactive molecules, particularly phenolic compounds and terpenes, two groups of allelochemicals that have been recognised for their allelopathic potential. Generally, essential oils have a higher toxicity than extracts, but if the extracts are obtained from plants known to produce toxic metabolites, the extracts may be more toxic than essential oils (Ferraz et al., 2022). There is a lack of comparative studies on both types of extracts, but published results indicate that EOs have a stronger growth inhibitory effect on weeds than PEs (Elghobashy et al., 2024). In addition, there is evidence that essential oils have the advantage of high biodegradability in the environment and relative safety for humans and other non-target organisms compared to synthetic pesticides (Giunti et al., 2022). The use of essential oils as biopesticides is associated with numerous challenges due to their inherent properties (lipophilicity and high volatility), production costs and manufacturing limitations (Gupta et al., 2023).
Ferraz, C. A; Pastorinho, M. R.; Palmeira-de-Oliveira, A.; Sousa, A.C.A. Ecotoxicity of plant extracts and essential oils: A review. Environ. Pollut. 2022, 292, 118319. https://doi.org/10.1016/j.envpol.2021.118319
Elghobashy, R. M.; El-Darier, S. M.; Atia, A. M.; Zakaria, M. Allelopathic Potential of Aqueous Extracts and Essential Oils of Rosmarinus officinalis L. and Thymus vulgaris L. J. Soil Sci. Plant Nutr. 2024, 24, 700-715. https://doi.org/10.1007/s42729-023-01576-x
Giunti, G; Benelli, G.; Palmeri, V.; Laudani, F.; Ricupero, M.; Ricciardi, R.; Maggi, F.; Lucchi, A; Guedes, R.N.C.; Desneux, N.; Campolo, O. Non-target effects of essential oil-based biopesticides for crop protection: Impact on natural enemies, pollinators, and soil invertebrates. Biological Control, 2022, 176:105071. https://doi.org/10.1016/j.biocontrol.2022.105071
Gupta, I.; Singh, R.; Muthusamy, S.; Sharma, M.; Grewal, K.; Singh, H.P.; Batish, D.R. Plant Essential Oils as Biopesticides: Applications, Mechanisms, Innovations, and Constraints. Plants, 2023, 12, 2916. https://doi.org/10.3390/plants12162916
-Lines 229–233: Rephrase for clarity. Suggest: “The regression parameters (B, C, D, and EC₅₀) indicated that seedling growth was more sensitive than seed germination in both species.”
We added an explanation about the sensitivity of measured parameters in the main text:
“Based on the calculated EC50 values for both EOs across all measured parameters, seedling growth (shoot and radical length) demonstrated greater sensitivity than seed germination” (lines 246-248).
-Figures: Make sure all figures have fully descriptive captions. Current ones are too brief (e.g., Fig. 3).
We clarified the captions of all the Figures.
-Lines 421–426: There is a bracket mismatch in the equation. Ensure the mathematical expression is correctly formatted.
We accepted your suggestion.
Y = C + (D − C) / 1 + exp (B (log X − log E))
-Reference Style: Ensure journal titles are consistently abbreviated or spelled out as per Plants guidelines.
We checked abbreviations of journal titles for all references in the manuscript.
Reviewer 2 Report
Comments and Suggestions for Authors
Authors must give background information in abstract
Authors must have antioxidant results in abstract
All results must indicate statistical ±SD
Conclusion is not clear
Introduction
Line 35-42 Only one citation is not sufficient
Line 66-78 this part needs revision. It is not clear “What authors want to explain here?”
Results
Phytochemical analysis must be shown first and then analysis. Authors must re -arrange manuscript.
The authors must show GCMS and UHPLC-DAD MS/MS chromatograms.
Also MS/MS data must be shown in main manuscript.
Author Response
Dear Reviewer 2,
Thank Reviewer 2 for the valuable corrections and all the suggestions about our paper. We have done our best to improve it following the remarks we received. All comments to amend the manuscript were accepted, and the manuscript was changed following suggestions. We did our best to clarify parts of the manuscript that needed clarification. Changes that we have made according to your suggestions are made throughout the manuscript and highlighted in a yellow background. Below are our point-to-point responses to the general as well as each specific comment and suggestion. In addition, English language editing and style polishing were done.
Reviewer 2
Comments and Suggestions for Authors
- Authors must give background information in abstract
- Authors must have antioxidant results in abstract
Thank you for your suggestion. We included background information and antioxidant values in the abstract.
- All results must indicate statistical ±SD
We included ±SD for all results.
-Conclusion is not clear
We rephrased and clarified the conclusion.
Introduction
-Line 35-42 Only one citation is not sufficient
As you suggested, we added two more citations.
Hossard, L.; Guichard, L.; Pelosi, C.; Makowski, D. Lack of evidence for a decrease in synthetic pesticide use on the main arable crops in France. Sci. Total Environ. 2017, 575, 152-161. https://doi.org/10.1016/j.scitotenv.2016.10.008
Gupta, I.; Singh, R.; Muthusamy, S.; Sharma, M.; Grewal, K.; Singh, H.P.; Batish, D.R. Plant Essential Oils as Biopesticides: Applications, Mechanisms, Innovations, and Constraints. Plants, 2023, 12, 2916. https://doi.org/10.3390/plants12162916
-Line 66-78 this part needs revision. It is not clear “What authors want to explain here?”
We appreciate your input. We clarified our goals in the Introduction.
Results
- Phytochemical analysis must be shown first and then analysis. Authors must re -arrange manuscript.
We rearranged the manuscript in the section of Results and first showed the phytochemical analysis.
The authors must show GCMS and UHPLC-DAD MS/MS chromatograms.
UHPLC-DAD chromatograms (recorded at 320 nm) of the extracts of two Artemisia species are depicted in Figure S1.
Also MS/MS data must be shown in main manuscript.
As you suggested, we added retention times, molecular ions (M-H-), and MS2 fragments (Q1 and Q3).

Reviewer 3 Report
Comments and Suggestions for Authors
The manuscript by Teodor Tojić et al., entitled 'Chemical screening and assessment of the allelopathic potential of Artemisia absinthium and A. vulgaris originating from Serbia', analyses the volatile, flavonoid and polyphenolic metabolite complexes of two species of wormwood, investigating their allelopathic effect on the germination of two problematic weed species in agriculture. The relevance of this topic is beyond doubt.
Overall, the work is of an excellent methodological and instrumental standard, and I would gladly recommend it for publication. However, I believe that the presentation of the results could be improved, and there are also a number of minor points that should be clarified before the manuscript can be accepted for publication.
I would also like to note separately the brevity of the text and the good structure of the presentation of the material and the results and their discussion. However, I would like to see a few lines about the prospects for the practical application of the results obtained by the authors in the Discussion or Conclusion section.
In metabolomics studies, correctly identifying compounds is critical, so I strongly recommend supplementing Table 1 with information on the M+ adduct and characteristic fragment masses of the MS² spectra to confirm the structure of each compound. It would also be useful to include a UV or total ion current chromatogram in the manuscript to demonstrate the quality of metabolite separation suitable for quantitative analysis. The identified metabolite should be labelled above each peak.
It should be explained on the basis of what data is available, which shows that isorhametin-3-O-glucoside and kaempferol-3-O-glucoside were identified specifically as glucosylated forms among other possible hexoses. If hexose analysis was not performed, the identification should be changed to 'kaempferol-3-hexoside', and so on.
Figures 2, 3 and 5 should be revised as there is no reliable evidence to support the extrapolation of the graph (red line) since the authors have no observations in the 0–0.25% range. The current version of the graphs looks very speculative!
In my opinion, the discussion lacks generalisations or hypotheses regarding how exactly chlorogenic acid and thujone can inhibit seed germination and seedling growth and which signalling or physiological pathways may be involved.
Minor comments:
In Section 4.3, parameters such as the ionisation energy of the mass spectrometric detector, the range of scanned masses and the method used to record the chromatogram (total ion current or SIM mode) should be added.
Author Response
Dear Reviewer 3,
Thank Reviewer 3 for the valuable corrections and all the suggestions about our paper. We have done our best to improve it following the remarks we received. All comments to amend the manuscript were accepted, and the manuscript was changed following suggestions. We did our best to clarify parts of the manuscript that needed clarification. Changes that we have made according to your suggestions are made throughout the manuscript and highlighted in a yellow background. Below are our point-to-point responses to the general as well as each specific comment and suggestion. In addition, English language editing and style polishing were done.
Reviewer 3
The manuscript by Teodor Tojić et al., entitled 'Chemical screening and assessment of the allelopathic potential of Artemisia absinthium and A. vulgaris originating from Serbia', analyses the volatile, flavonoid and polyphenolic metabolite complexes of two species of wormwood, investigating their allelopathic effect on the germination of two problematic weed species in agriculture. The relevance of this topic is beyond doubt.
Overall, the work is of an excellent methodological and instrumental standard, and I would gladly recommend it for publication. However, I believe that the presentation of the results could be improved, and there are also a number of minor points that should be clarified before the manuscript can be accepted for publication.
-I would also like to note separately the brevity of the text and the good structure of the presentation of the material and the results and their discussion. However, I would like to see a few lines about the prospects for the practical application of the results obtained by the authors in the Discussion or Conclusion section.
Thank you for your suggestion. We added a few lines in the Conclusion section.
-In metabolomics studies, correctly identifying compounds is critical, so I strongly recommend supplementing Table 1 with information on the M+ adduct and characteristic fragment masses of the MS² spectra to confirm the structure of each compound. It would also be useful to include a UV or total ion current chromatogram in the manuscript to demonstrate the quality of metabolite separation suitable for quantitative analysis. The identified metabolite should be labelled above each peak.
The new table (Table 1) has now been added to the revised version of the manuscript. It includes retention times, molecular ions (M-H-), and MS2 fragments (Q1 and Q3) used for confirmation of the compounds and quantification. The total ion current (TIC) chromatogram cannot be displayed under these LC-MS conditions (SRM experiment), while UV (DAD) chromatograms are depicted in the revised version of the manuscript. Please note that in this experiment, only the MS detector (SRM transitions) was used for quantification, so no attention was paid to DAD separation. very good separation and isolation of individual compounds were obtained with MS transitions (Table 1). Namely, DAD chromatograms show many more compounds than we were able to quantify with available standards. So we cannot annotate all peaks, and the largest peak at 6.74 min should be isorhametin 3-O-rutinoside, which was also quantified.
-It should be explained on the basis of what data is available, which shows that isorhametin-3-O-glucoside and kaempferol-3-O-glucoside were identified specifically as glucosylated forms among other possible hexoses. If hexose analysis was not performed, the identification should be changed to 'kaempferol-3-hexoside', and so on.
We understand your concern, and we otherwise identify the unknown glucosylated forms as hexoses exactly as you suggested. But in this work, only quantification was performed using available analytical standards, so the nature of the glycosides was known in advance.
-Figures 2, 3 and 5 should be revised as there is no reliable evidence to support the extrapolation of the graph (red line) since the authors have no observations in the 0–0.25% range. The current version of the graphs looks very speculative!
As the reviewer noted, we have changed the scale on the x-axis and hope that the graphs are now correct.
-In my opinion, the discussion lacks generalisations or hypotheses regarding how exactly chlorogenic acid and thujone can inhibit seed germination and seedling growth and which signalling or physiological pathways may be involved.
We agree with your proposal.
Xu et al. (2025) discovered DHAR1 (dehydroascorbate reductase 1), an enzyme responsible for ascorbate regeneration in plants, as the target site of chlorogenic acid, whose inhibition of activity led to the accumulation of H2O2 and the reduction of proteins involved in water transport and photosynthesis.
Xu, J.; Chen, L.; Wang, S.; Zhang, W.; Liang, J.; Ran, L.; Deng, Z.; Zhou, Y. Chemoproteomic Profiling Reveals Chlorogenic Acid as a Covalent Inhibitor of Arabidopsis Dehydroascorbate Reductase 1. J. Agric. Food. Chem. 2025, 73, 908-918. https://doi.org/10.1021/acs.jafc.4c07955
The complex chemical composition of essential oils makes it difficult to isolate their components and determine their potential mode of action. The phytotoxic effects of essential oils have been related to plant growth reduction, leaf chlorosis, changes in plant cells, mitosis, cellular respiration, chlorophyll content, membrane depolarization and ion leakage, cuticular waxes, oxidative stress, and microtubular polymerization (Raveau et al., 2020).
Raveau, R.; Fontaine, J.; Sahraoui, A. L. Essential Oils as Potential Alternative Biocontrol Products against Plant Pathogens and Weeds: A Review. Foods 2020, 9, 365. https://doi.org/10.3390/foods9030365
Minor comments:
-In Section 4.3, parameters such as the ionisation energy of the mass spectrometric detector, the range of scanned masses and the method used to record the chromatogram (total ion current or SIM mode) should be added.
We added all the information.

Round 2
Reviewer 2 Report
Comments and Suggestions for Authors
Accept in present form
Reviewer 3 Report
Comments and Suggestions for Authors
The authors have clarified all the controversial points and reflected them not only in the response to the review, but also in a substantially revised version of the manuscript.
The illustrations have also been modified in accordance with the comments made.
The chromatograms presented by the authors indicate good analytical technique and chromatographic separation.
In my opinion, the manuscript can be published in its current form.